# Dynamic Monitoring and Ecological Risk Analysis of Lake Inundation Areas in Tibetan Plateau

Dongchuan Wang [1,2], Hua Chai [2,*], Zhiheng Wang [1,2], Kangjian Wang [2], Hongyi Wang [2], Hui Long [2], Jianshe Gao [2], Aoze Wei [2] and Sirun Wang [2]

1    Tianjin Key Laboratory of Soft Soil Characteristics and Engineering Environment, Tianjin Chengjian University, Tianjin 300384, China
2    School of Geology and Geomatics, Tianjin Chengjian University, Tianjin 300384, China
*    Correspondence: chaihua666@gmail.com

**Abstract:** Lake inundation is one of the most important hydrological factors affecting lake ecosystems. In order to accurately and timely grasp the spatio-temporal pattern of the lake inundation area, and reveal the ecological evolution of the lake landscape, this paper quantifies the inundation dynamics of lakes on the Tibetan Plateau in the past 20 years and analyzes the spatio-temporal characteristics of the inundation area from four aspects: the region, type, altitude and recharge mode of the lake. Combined with the water inundation frequency, the landscape inundation frequency index is constructed and applied to the landscape ecological risk index to explore the spatio-temporal dynamic changes of landscape ecological risk in the inundation area. The results show that the change of the lake-inundated area first decreases and then increases in 2000–2020, the salt lakes and low-altitude lakes have the largest inundation areas, accounting for 83.2% and 55.6% of the total inundated area, respectively; the change intensity of lake inundation frequency is relatively large, and the alternate changes of the lake water–land junction area are enhanced, and the area of permanent lake increases; inundation has the greatest impact on bare land and grassy landscapes; the study area is dominated by lower-risk and lowest-risk areas, accounting for 84.9% of the total area of risk areas, but most areas are transformed from lower-lowest risk to medium-higher risk. This study provides a case of dynamic monitoring of lake inundation areas, which is helpful to formulate ecological restoration and risk prevention measures in lake inundation areas, and can also be used for ecological risk research in similar areas.

**Keywords:** inundated area; inundation frequency; ecological risk; landscape pattern; Tibetan Plateau





## 1. Introduction

In the lake ecosystem, the change of hydrological characteristics is of great significance to the quality and biodiversity of the entire ecosystem [1,2]. The frequency of flooding reflects the length and times of water flooding, which is considered to be the most important hydrological factor affecting the lake ecosystem [3]. Comprehensive analysis of the temporal and spatial variation characteristics of inundation frequency in lake inundation areas reveal that the impact of inundation frequency on landscape changes is of great significance for studying the inundation risk brought about by lake expansion. In recent decades, the Tibetan Plateau has experienced obvious climate change, resulting in strong temporal and spatial differences in lake changes in this region [4–6]. The vast majority of lakes show an obvious expansion trend, which is manifested as the increase of lake area [7], the rise of water level [8] and the increase of water quantity [9]. These changes lead to the expansion of lake flooded areas and a series of impacts. In 2018, China issued 'Guidance on the Implementation of Lake Chief System in Lakes', which focused on lake ecological protection and was of great significance to maintain lake ecological security.

At present, there have been a lot of studies on lake monitoring on the Tibetan Plateau. Wan et al. [10] conducted statistical analysis on the number and area of lakes on the Tibetan

Plateau, Chen et al. [11], studied the impact of lake water level changes on the Tibetan Plateau and Jing et al. [12] quantified the main driving factors of lake change on the Tibetan Plateau. These studies focus on lake change monitoring and the driving factors of lake change, providing important references for related research. However, to fully understand the changing characteristics of plateau lakes, lake inundation is an important factor affecting lake ecosystems, and it is necessary to carry out dynamic monitoring. Cheng et al. [13] evaluated the impact of the rapid expansion of Lake Angzi Co on adjacent villages. Chen and Wang et al. [14,15] have shown that the level of the Selin Co Lake has been rising in recent years, leading to the inundation of surrounding high-quality grasslands, and the decline of animal husbandry has had a serious impact on local herdsmen. Although Liu and Wu et al. [16,17] have recognized that inundation is an important factor affecting the landscape changes around the lake, there is still a lack of systematic and quantitative research on how the inundation process of different characteristics affects the landscape changes in the inundated area. Therefore, this paper conducts dynamic monitoring of a lake inundation area and landscape in the inundated area, and quantitatively studies the response of the landscape to the inundation process.

The study of landscape ecological risk assessment has gradually developed from single evaluation to comprehensive evaluation, and the evaluation models and methods have been continuously improved [18]. In recent years, domestic and foreign scholars pay attention to discussing the evaluation objectives and methods of landscape ecological risk assessment and build relevant research systems [19]. There are two main models in the evaluation method. One is based on the "risk source sink", that is, the regional landscape ecological risk assessment is carried out through the "risk source analysis—risk receptor identification—exposure and hazard assessment" [20]. The other is based on the landscape index method to quantitatively evaluate and describe the overall quality and spatial-temporal differentiation characteristics of regional ecological risk [21,22]. These two evaluation methods have been widely used in the research scope of watersheds [23,24], administrative units [25,26], mining areas [27] and nature reserves [28,29]. The former approach focuses on the identification of risk sources and risk receptors. In the evaluation process, the determination of the risk sources and each index is subjective and has a great influence on the accuracy of the evaluation results. The latter risk approach is widely applicable as it takes account of scale effects, temporal changes, and regional spatial heterogeneity. The approach also facilitates spatial visualization of results for decision making, regional risk prevention, optimization and management of regional landscape patterns [30].

In the application of the landscape index method, common landscape pattern indexes include fragmentation, vulnerability, dominance [31,32] and shape index [33]. Forbes, V.E, Mo and W [34,35] used the two indicators of landscape disturbance and landscape vulnerability to quantify the ecological risk, the product of these two indicators was used to estimate the potential ecological loss, and then they calculated the risk value of the region combined with the risk probability. However, some studies have pointed out that the method has certain limitations [36]. A reliable quantitative method is the premise of analyzing and managing ecological risk. The key steps of the existing methods are too simplified, that is, the vulnerability is measured as a static constant by the expert scoring method, which is certainly subjective and it is difficult to evaluate its dynamic characteristics [37,38]. In fact, the ecosystem is a dynamic continuum, characterized by varying degrees of vulnerability according to external pressure sources and internal characteristics. The method that only considers the static differences between landscape types cannot fully reflect the temporal and spatial heterogeneity of risk, especially the long-term heterogeneity. Therefore, this method still needs to be improved in practical application. Therefore, this paper constructs the landscape inundation frequency index as a dynamic indicator to quantify the landscape vulnerability and conducts the spatiotemporal dynamic analysis of ecological risk in lake inundation areas.

This paper uses long-term water data and land use data to dynamically monitor the lake inundation area and the landscape in the inundated area from four aspects: lake region, type, altitude and replenishment method, and quantitative research on the response of the landscape to the inundation process. On this basis, the ecological risk analysis of the lake inundation area is carried out, and the landscape inundation frequency index is constructed as a dynamic index to quantify the landscape vulnerability index, from the perspective of time and space. The risk changes of the lake-inundated area were analyzed to provide a reference for ecological restoration and risk prevention in the area around the lakes on the Tibetan Plateau.

## 2. Materials and Methods

### 2.1. Study Area

The Tibetan Plateau (26°00′–39°47′ N, 73°19′–104°47′ E) is the largest and highest plateau in China, which is known as the "roof ridge of the world". From the south of the Himalayas to the north of the Kunlun Mountains, Altun Mountains and Qilian Mountains, the west is the Pamir Plateau and Karakorum Mountains, and the east and northeast are connected with the western part of Qinling Mountains and the Loess Plateau. There are various types of lakes on the Tibetan Plateau, which has more than 1500 lakes, with a total area of 42,816.10 km², accounting for about 49.5% of the total lake area in China. They are mainly distributed in the western region of the plateau.

Studies have shown that the lake area on the Tibetan Plateau has increased significantly, and about three-quarters of the alpine lakes have dramatically expanded since the late 1990s [39,40]. Especially after 2000, the lakes expanded rapidly; lake expansion has become one of the most outstanding environmental change events on the Tibetan Plateau, and the impact of small lakes on the surrounding environment is very limited [41]. This paper selects 52 representative lakes with an area greater than 50 km² from 2000 to 2020 as the research objects. In order to facilitate discussion, the Tibetan Plateau is divided into four regions based on mountains and terrain factors, as shown in Figure 1.

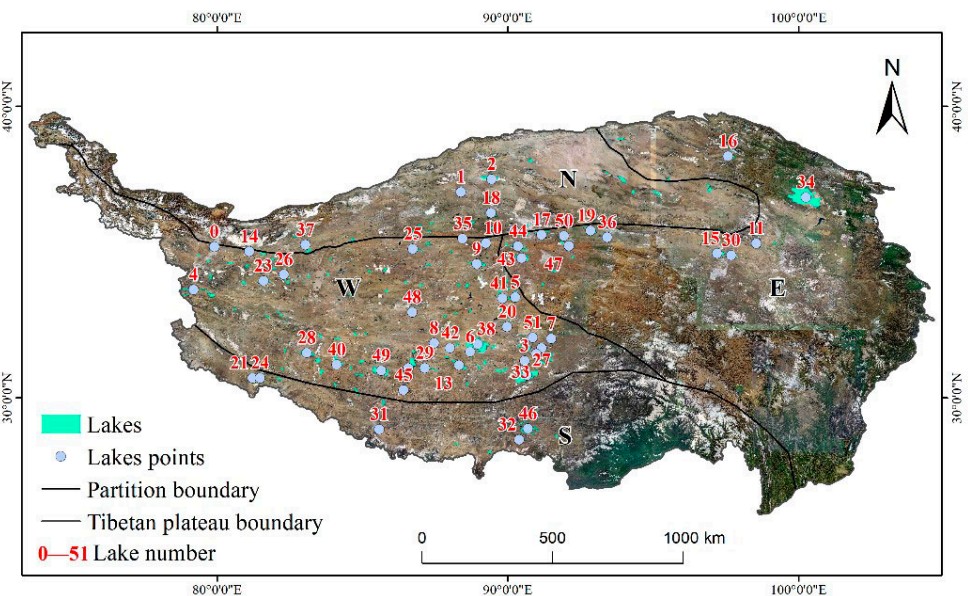

**Figure 1.** Location of the study area.

### 2.2. Data Sources and Processing

The water data (Joint Research Centre Monthly History V1.3 dataset) were obtained based on the Google Earth Engine platform with the resolution of 30 m. The dataset contains the location distribution of surface water from 1984 to 2020 [42]. From May to October is the non-icing period of the Tibetan Plateau, and the precipitation is concentrated [43], so

the data from May to October from 2000 to 2020 in this dataset were selected. We exported the water data and manually deleted the non-lake elements, and the lake was used as the unit for the maximum range synthesis of lake flooding (that is, pixels that were marked as water at least once). We obtained GF-1 images from 2016 and 2020 with the resolution of 8 m (http://ids.ceode.ac.cn/ (accessed on 22 September 2022)).

Land use data were obtained from the Land Cover Database (www.globallandcover.com (accessed on 14 May 2022)) with a resolution of 30 m. The data include a total of 10 first-level types, namely, cultivated land, forest land, grassland, shrub land, wetland, bare land, water, tundra, artificial surface, glaciers and permanent snow cover [44]. The Tibetan Plateau elevation data and the Tibetan Plateau glacier distribution data were obtained from the Tibetan Plateau Data Center (http://data.tpdc.ac.cn/zh-hans (accessed on 14 May 2022)).

*2.3. Research Methods*

2.3.1. Lake Inundation Area Division

The lake-inundated area refers to the area that was judged to have appeared as water during the study period, except for the area of the perennial stable lakes [45]. The inundation frequency refers to the proportion of the inundated time of a certain area to the total inundated time during the study period. Based on the maximum inundation range of lakes, this paper uses monthly data to calculate the annual inundation frequency and divides the inundation areas of various lakes [46]. The calculation formula is as follows:

$$F(y) = \frac{\sum_{i=1}^{n} D_i \times t}{N} \times 100\% \tag{1}$$

In the formula, *F(y)* represents the annual inundation frequency of year *y*; *n* represents the total number of months of water data in year *y*; $D_i$ is the pixel value of the distribution of water in the *i*-th month of year *y* (water is 1, non-water is 0); *t* is the number of days between adjacent months; *N* is the total number of days in year *y*. According to previous research and combined with the actual situation of the study area [47], due to the slight errors in the water data in some areas, this paper regards $0\% < F \leq 30\%$ as a low inundation frequency area, and $30\% < F \leq 60\%$ as a medium inundation frequency area, $60\% < F \leq 90\%$ is a high inundation frequency area and $F > 90\%$ is permanent water.

2.3.2. Accuracy Verification

Confusion matrix and kappa coefficient were used to test the reliability of classification results [48]. In this paper, accuracy verification was carried out with samples from field investigations around the lakes and randomly selected points on the synchronous GF-1 imageries. The GF-1 images from 2016 and 2020 were visually interpreted, and lakes with large inundation area in the study area were selected for comparison and verification of spatial distribution.

2.3.3. Evaluation Unit Division of Lake Flooded Area

The Tibetan Plateau has a large number of lakes, and the changes of hydrological elements in different lakes have great differences [49]. This paper analyzes the inundated area from the four aspects of lake division, nature, altitude and replenishment method. The division criteria are as follows: The Tibetan Plateau is divided into east (E), south (S), west (W) and north (N) regions based on mountains and terrain factors [50]; the mineralization degree of lakes was determined according to the data of "Chinese Lake Chronicles" [51], "China Lake Distribution Atlas" [52]; based on the mineralization degree of lake water, it is divided into salt lakes (greater than or equal to 35 g/L), saltwater lakes (between 1 g/L and 35 g/L) and freshwater lakes (less than or equal to 1 g/L) [53,54]; this paper defines greater than or equal to 3000 m and less than 4500 m as a low altitude area, greater than or equal to 4500 m and less than 5000 m as a medium-altitude area and greater than or equal

to 5000 m is a high-altitude area; recharge is divided into ice-snow melt water recharge, surface runoff recharge, river recharge and others [40].

### 2.3.4. Landscape Inundation Frequency Index

This paper refers to the research of Fan Deng [55] and combines the characteristics of lake inundation area to construct a landscape inundation frequency index. Considering the sensitivity of landscape types after being disturbed by inundation, the lake inundation frequency was divided into 10 groups at 10% intervals, and the response of each landscape to different inundation frequencies was quantitatively studied. The formula is as follows:

$$V_i = \frac{1}{c_i} \times \sqrt{\frac{\sum_{i=1}^{n} b_{ji} - c_i}{n}} \tag{2}$$

In the formula, $V_i$ is the landscape inundation frequency index of the $i$-th type of landscape; $b_{ji}$ is the area proportion of landscape type $i$ in the $j$-th group; $c_i$ is the proportion of the $i$-th type of landscape area to the total area of the inundated area; $n$ is the total number of inundated frequency groups.

Sensitivity is an intrinsic property of the ecosystem and it is defined as the degree to which the system is affected by those perturbations [38]. Vulnerability is conceptualized as susceptibility to exposure to perturbations or external stresses, sensitivity to perturbation and a lack of adaptive capacity; vulnerability increases with sensitivity, making the structure and functions of the landscape more susceptible to change by external disturbances [56]. The higher the value of $V_i$, the greater the sensitivity of the landscape to the inundation frequency, and the more fragile the landscape is.

### 2.3.5. Construction of Ecological Risk Index in Inundated Area

Landscape ecological risk depends on the strength of the external disturbance and the internal resistance of the ecosystem in the study area [57]. This paper establishes the relationship between land use types and ecological risks from two aspects. One is to learn from the previous research results and introduce the landscape disturbance index; the other is to apply the landscape inundation frequency index to the landscape vulnerability index, using the ecological risk assessment. The relationship between the area of different landscape types in the unit and the inundation frequency was used to quantitatively analyze the vulnerability of the landscape. Therefore, a landscape ecological risk assessment model is constructed to analyze the temporal and spatial distribution and dynamic changes of ecological risks. The calculation formula of the landscape ecological risk index is as follows:

$$ERI_k = \sum_{i=1}^{n} \frac{A_{ki}}{A_k} S_i V_i \tag{3}$$

In the formula, $ERI_k$ is the ecological risk index of the $k$-th ecological risk community; $A_{ki}$ is the area of landscape type $i$ in the $k$-th risk community; $A_k$ is the area of the $k$-th risk community; $S_i$ is the landscape disturbance index of the $i$-th landscape; $V_i$ is the landscape inundation frequency index of the $i$-th type of landscape; and $N$ is the number of landscape types.

(1) Landscape disturbance index.

The degree of landscape disturbance is used to describe the degree of disturbance of different landscape types by human activities, and to describe the degree of external disturbance of the landscape in the inundated area [58]. We chose landscape fragmentation, landscape separation and landscape dominance to quantitatively analyze the disturbance degree of the inundated area. The calculation formula is as follows:

$$S_i = a_1 C_i + a_2 N_i + a_3 D_i \tag{4}$$

In the formula, $S_i$ is the landscape disturbance index, $a_1$, $a_2$ and $a_3$ are the weight of the corresponding index and $a_1 + a_2 + a_3 = 1$. According to the actual situation of the study area and relevant literature research, the values were assigned as 0.5, 0.2 and 0.3, respectively [59]. Landscape fragmentation ($C_i$) describes the fragmentation degree of each land use type, which reflects human modification; higher values indicate greater landscape disturbance. Landscape separation ($N_i$) reflects the degree of separation or isolation between land use patches. Land use types with a higher degree of separation are characterized as more dispersed and complex in their geographical distribution. Landscape dominance ($D_i$) indicates the degree of influence of a given land use type on the landscape. For calculation, please refer to [60].

(2) Landscape vulnerability index.

Landscape vulnerability is used to represent the sensitivity of different landscape types after being disturbed by the outside [61]. Most of the previous studies used the subjective weight of ecological risks of land use, coupled with the area ratio of the corresponding land use type to assign values [62], which has a relatively large subjective value. In this paper, the constructed landscape inundation frequency index is applied to the calculation of landscape vulnerability index (Formula (2)), quantifying the dynamic changes in the vulnerability of various landscapes within a lake inundation area.

### 2.3.6. Ecological Risk Assessment Unit and Ecological Risk Classification

In order to express the spatial heterogeneity of the landscape pattern and spatially display the ecological risk index of the regional landscape, it is necessary to divide the study area into evaluation units [63]. According to the research of landscape ecology, the optimal range of landscape sample area selection is generally 2–5 times the average patch area [64]. Based on the full consideration of landscape spatial heterogeneity, patch size and lake-inundated area, using ArcGIS software, the equal square grid of 1.5 km × 1.5 km was used as a small evaluation unit for equal interval sampling. In data processing, grids are used as small study units for space sampling; taking the geometric center of the grid as the sampling point of the landscape ecological risk index, the landscape ecological risk index of each community was calculated, and the spatial distribution map of the landscape ecological risk in the whole study area was obtained by the interpolation method. Using the natural breakpoint method, the landscape ecological risk was divided into five levels [65], as shown in Table 1.

**Table 1.** Ecological risk classification.

| Risk Level | Lowest Risk | Lower Risk | Medium Risk | Higher Risk | Highest Risk |
|:---:|:---:|:---:|:---:|:---:|:---:|
| ERI | <0.272 | 0.272–0.459 | 0.459–0.789 | 0.789–1.314 | >1.314 |

In the calculation process, we used the ArcGIS and Fragstats software to build a cross-platform, process-based mass calculation method of the landscape ecological risk index. The computation capacity of this method exceeds 10,000 grids simultaneously and enables computations for large study areas where the workload increases exponentially, which makes the calculation process precise and process-based and provides a reproducible operation mode for related research (Figure 2).

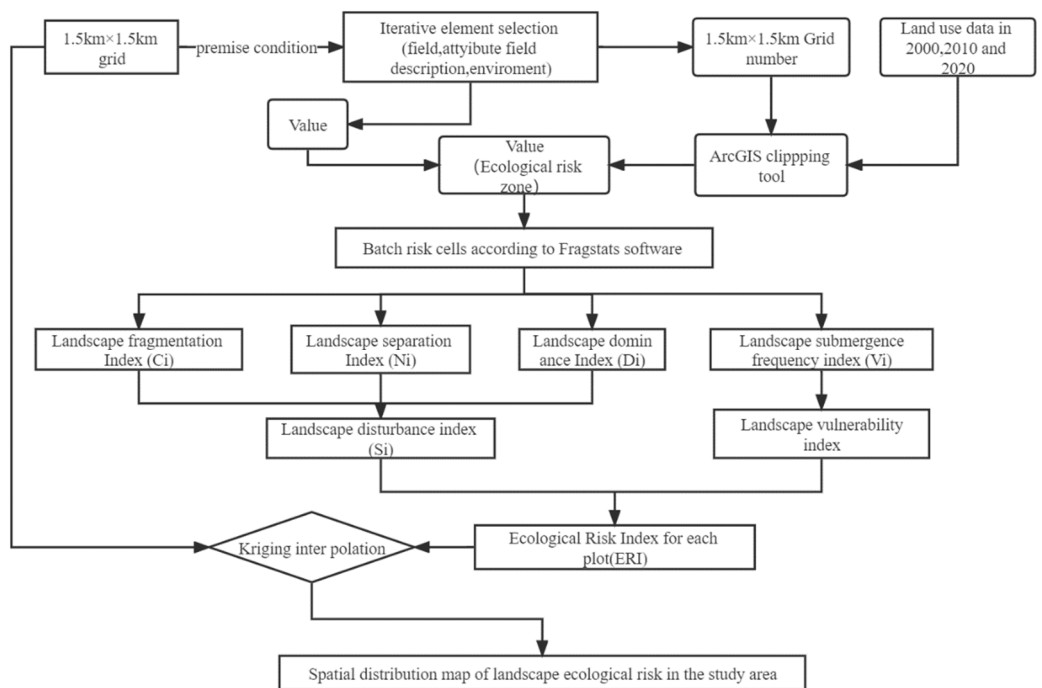

**Figure 2.** Flowchart for calculating the landscape ecological risk index.

## 3. Results

### 3.1. Verification of Extraction of the Lake Inundation Area

Within the research area, random points, together with parts of the field investigation sample points, altogether 525 verified sample points (421 water sample points and 104 non-water sample points), were selected to verify the accuracy of the extraction. It showed that the results met the accuracy requirements with the kappa coefficient of 0.81. What is more, results from typical lakes such as Selin co Lake, Dogaicoring Qangco Lake, Xijir Ulan Lake and Ayakkum Lake were selected to compare with the visual interpretation results from GF-1 imageries. As shown in the Figure 3, the bottom picture shows the 2020 GF-1 image data, the yellow line is the inundation boundary obtained in this article, the red line is the manual interpretation to obtain the GF-1 2020 lake boundary and the green line is the GF-1 2016 lake boundary, making a spatial comparison between the inundation area of the lake and the lake boundary. The average error is verified within one pixel.

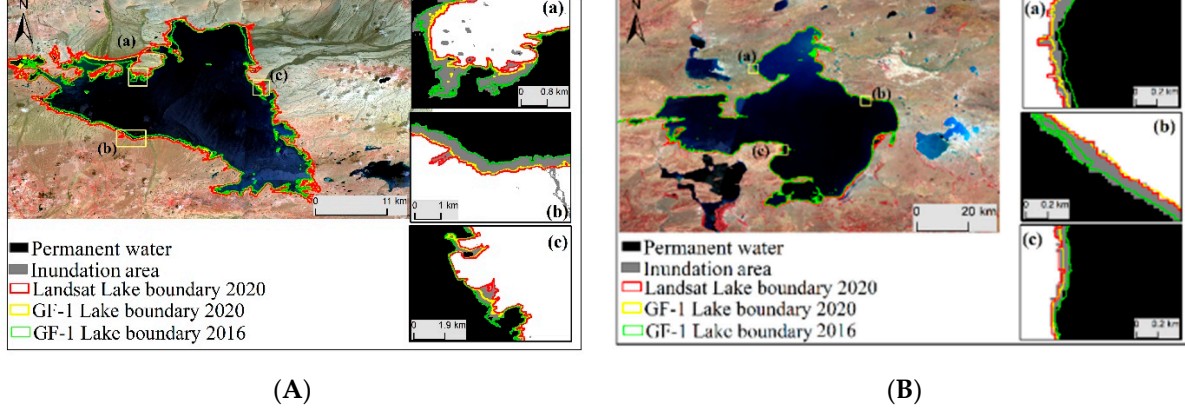

**(A)**                                                                 **(B)**

**Figure 3.** *Cont.*

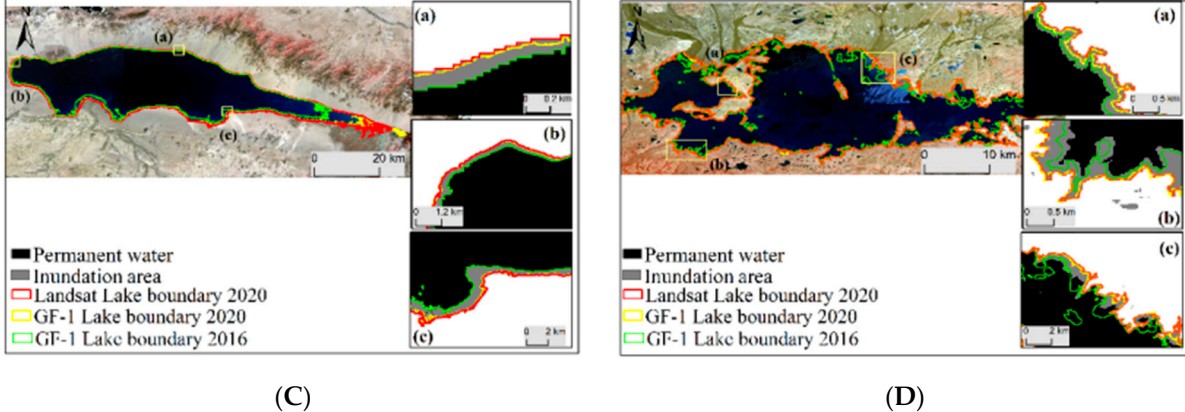

(**C**)          (**D**)

**Figure 3.** Comparison of spatial distribution of inundated areas of lakes (**A**) Dogaicoring Qangco Lake (a) Western region (b) Southern region (c) Eastern Region; (**B**) Selin co Lake(a) Northern region (b) Eastern region (c) Western Region; (**C**) Ayakkum Lake (a) Northern region (b) Western region (c) Eastern Region; (**D**) Xijir Ulan Lake (a) Eastern region (b) Northern region (c) Western Region.

### 3.2. Variation Characteristics of Lake-Inundated Area

Changes in the area of inundated areas of various lakes on the Tibetan Plateau are observed (Figure 4), and 31 are inundated in areas ranging from 0 to 100 km², accounting for 54% of the total number of lakes. There are 13 lake-inundated areas in the range of 100–200 km², 4 lake-inundated areas in the range of 200–300 km² and 3 lake-inundated areas above 300 km². It can be seen that there are significant differences in the inundation changes of different lakes.

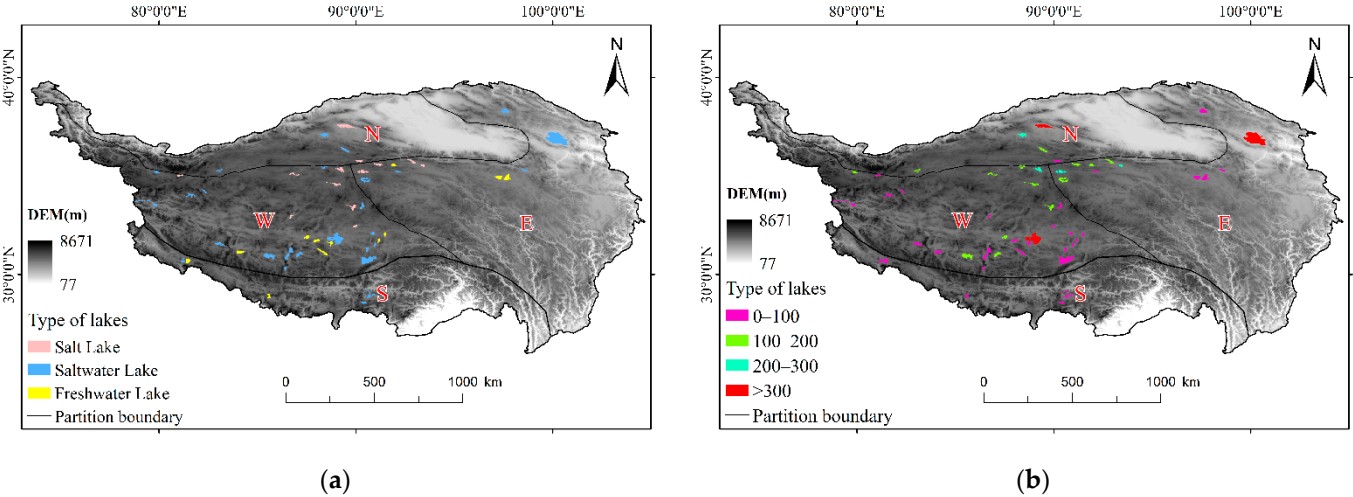

(**a**)          (**b**)

**Figure 4.** Change of Inundated Area in Tibetan Plateau from 2000 to 2020. (**a**) Inundated Area; (**b**) Type of lakes.

Based on the lake inundation information from 2000 to 2020, the variation characteristics of lake inundation areas in Tibetan Plateau are analyzed from four aspects (Figure 5):

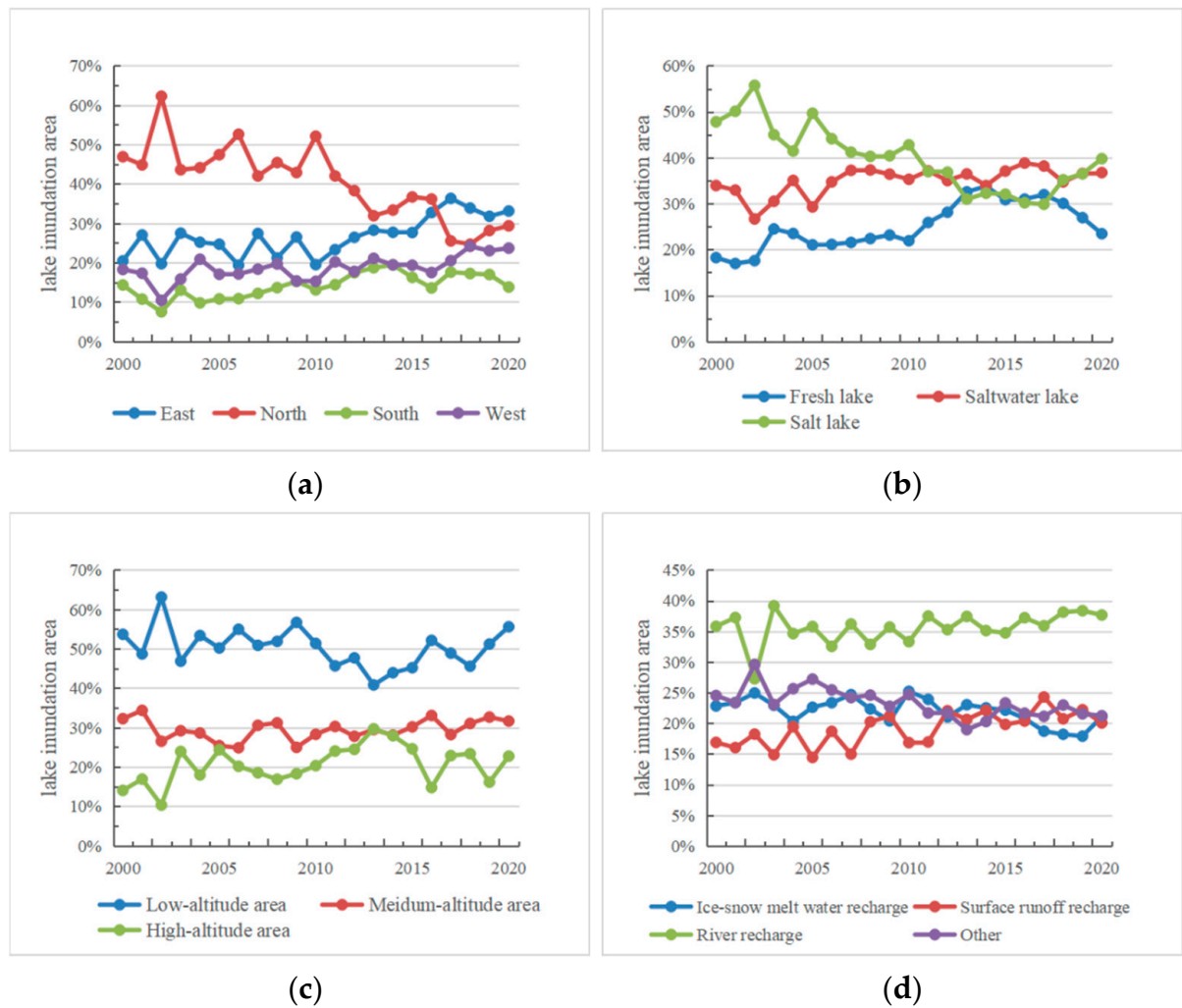

**Figure 5.** The area proportion of Lake Inundation area in Tibetan Plateau. (**a**) The lake region; (**b**) The type of lakes; (**c**) The lake altitude; (**d**) The replenishment mode.

From the area where the lakes are located, 54% of the lakes are concentrated in the western region. The lake density in this region is greater than that in other regions, but the inundated area in the western region is smaller than that in the northern and eastern regions. From 2000 to 2020, the inundated area of each region fluctuated. The northern region had the largest inundated area of 302.58 km$^2$, accounting for 52.8% of the total inundated area. Among them, the inundated area of Ayakumu Lake had the largest change of 240.25 km$^2$, accounting for 79.48% of the inundated area in the northern region, and the inundated area is second only to Selin Co Lake in the western region; the eastern inundated area is second only to the northern region, accounting for 25.7% of the total inundated area; the western region accounted for 15.2% of the total inundated area, of which the inundated area of Selin Co Lake was the largest; the change of the inundated area in the southern region was relatively stable for 20 years.

From the perspective of lake type, the inundated areas of salt lakes and saltwater lakes are large, accounting for 83.2% of the total inundated area from 2000 to 2020. Among them, the saltwater lake Selin Co Lake has the largest inundated area, followed by the salt lake Ayakum Lake. The total inundated area of the two lakes is about 556.7 km$^2$; the inundated area of freshwater lakes has changed relatively steadily, and the lake area has decreased from 2016 to 2020, accounting for 34.7% of the total inundated area.

In terms of altitude range from 2000 to 2020, the inundated area in the low-altitude area was the largest, accounting for 58.3% of the total inundated area; the change in the

inundated area in the middle-altitude area was relatively stable; the inundated area in the high-altitude area increased at a uniform rate from 2009, and it was equal to the area of the medium-altitude area in 2014.

In terms of the replenishment mode, the lakes supplied by rivers have the largest inundated area, accounting for 36% of the total inundated area; the inundated area of lakes replenished by ice-snow melt water shows an increasing trend, accounting for 24% of the total inundated area, with greater fluctuations from 2001 to 2005. Among them, the glaciers in the eastern section of Nyainqentanglha Mountain, where lakes are concentrated, and the glaciers in the southern part of the Qiangtang Plateau have retreated significantly, which have a greater driving effect on the expansion of lakes such as Dorsuodongcuo Lake and Cuona Lake. The area of the lake-inundated area recharged by surface runoff changed a little, accounting for 18% of the total inundated area.

### 3.3. Analysis of Lake Inundation Frequency and Landscape Change of Lake Inundation Area

#### 3.3.1. Variation of Inundation Frequency

The inundated frequency of lakes in different regions shows obvious differences (Figure 6). From 2000 to 2020, the area of low-frequency inundation areas showed a downward trend as a whole, from 38.7% to 20.8%. Among them, the northern region has the most obvious decline, from 11.6% to 2.18%, and the southern region is relatively stable; the area of the medium-frequency inundation area showed fluctuation changes, with a downward trend from 2000 to 2015, and an increase from 2016 to 2020; the area of the high-frequency inundation area showed a downward trend, from 33.5% to 21.3%, but the proportion of the eastern area increased; the area of permanent water is on the rise as a whole, rising from 23.9% to 26.3%; the eastern area has decreased, and the southern area has less change. Among them, the eastern region has declined compared with other regions, and the southern region has less changes. In recent years, in the Tibetan Plateau, the intensity of fluctuations in the inundation frequency is relatively large, indicating that the alternate changes of the land–water boundary area have enhanced, and the permanent water has grown steadily, indicating that the lake has been expanding in recent years.

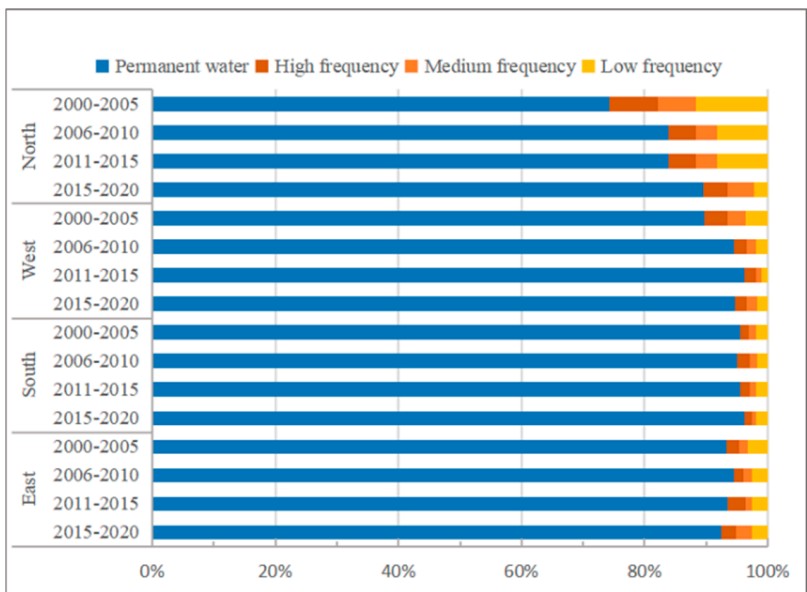

**Figure 6.** The proportion of different inundated areas from 2000 to 2020.

According to the lake inundation frequency, the range of various inundation areas is divided (Figure 7). Due to the large scale of the Tibetan Plateau, the specific lakes are not clear in the large scale, so the areas with larger inundated areas and dense lakes in the four regions of the Tibetan Plateau were selected to be enlarged as Figure 7a–d.

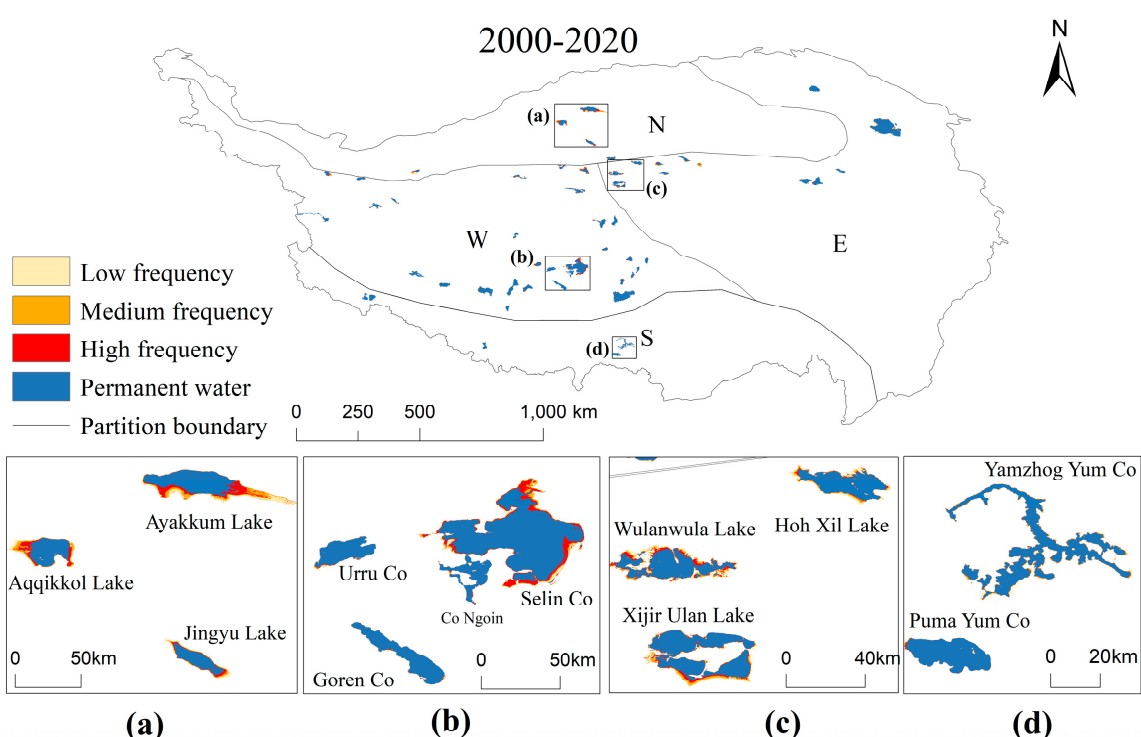

**Figure 7.** The distribution of various lake inundations from 2000 to 2020. (**a**) Lakes of the northern region; (**b**) Lakes of the Western region; (**c**) Lakes of the southern Region; (**d**) Lakes of the eastern region.

From a spatial perspective, there are many high mountains in the northern region, with a sufficient glacial melt water supply, and the proportion of permanent water and moderately inundated areas has increased significantly, which shows that the lakes in this area show a rapid expansion trend and the alternate changes of the inundated area are enhanced. The areas of various inundated areas in the southern region have little change, indicating that the lake water in the southern region is relatively stable; the area of the high-frequency inundation area in the eastern region has increased, indicating that the water and land around the lake area in this region are alternated frequently; the lakes in the western region are dense and the proportion of permanent water has increased, indicating that the lakes in this region have a trend of expansion.

### 3.3.2. Landscape Changes in the Inundated Area

According to the land use type data, the overall distribution and changes of land types in the study area in 2000, 2010 and 2020 were obtained (Table 2, Figure 8). The main landscape types in the lake-inundated area except water are grassland and bare land, accounting for 85.57% of the total area of the inundated area. The landscape around salt lakes is dominated by bare land, and the saltwater and freshwater lakes are dominated by grassland. The land use in the study area has changed most significantly from 2000 to 2020. The bare land is mainly transformed into water and wetland, with an area of 1664 km$^2$ and 87.38 km$^2$, respectively, accounting for 78.6% and 34.60% of the transformed area. From 2000 to 2020, the water continued to expand and advance towards medium-low inundation areas. The land use map and the inundation frequency map are superimposed to calculate the proportion of various landscapes in each inundation frequency area. In 2000, 98.2% of the water was distributed in the permanent water area, and in 2020, it dropped to 88.9%, and the water landscape continued to expand to the medium- and low-frequency inundation areas. In 2000, except for cultivated land and water, other landscapes were evenly distributed in each inundated area. From 2010 to 2020, with the continuous

expansion of water, the landscape in high-frequency inundation areas decreased rapidly, and the main distribution areas were in the medium-low-frequency inundation areas.

**Table 2.** Various types of land use area (km$^2$) and change rate (%) in 2000–2020.

| Type | 2000 | 2010 | 2020 | Change Rate (2000–2020) |
|---|---|---|---|---|
| Water | 24,360.82 | 26,444.18 | 27,750.96 | 13.92 |
| Grass land | 2107.12 | 1275.63 | 757.19 | −64.07 |
| Bare land | 2134.79 | 988.11 | 511.28 | −76.05 |
| Wetland | 373.71 | 406.35 | 203.35 | −45.59 |
| Shrub | 329.91 | 194.56 | 85.16 | −74.19 |
| Woodland | 4.97 | 2.5 | 0.43 | −91.35 |
| Arable land | 0.74 | 0.74 | 3.74 | 405.41 |
| Artificial land | 0.19 | 0.2 | 0.16 | −0.03 |

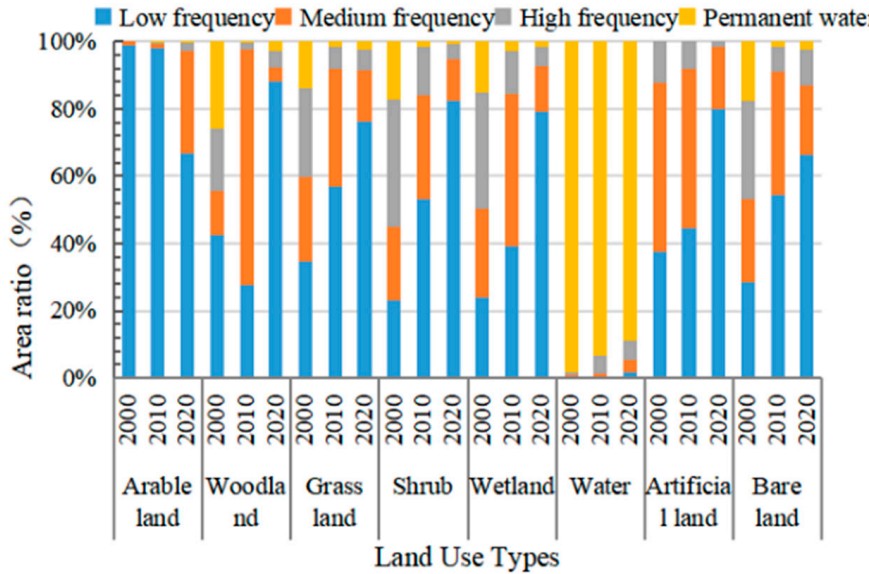

**Figure 8.** The proportions of various land use types.

*3.4. Analysis of Temporal and Spatial Changes of Ecological Risk in Lake-Inundated Areas of the Tibetan Plateau*

3.4.1. Change Analysis of Landscape Pattern Index

Based on Fragstats and Excel software, the landscape pattern index of various land use types in 2000, 2020 and 2020 was obtained, and further analysis of its change law (Table 3) can be seen. From 2000 to 2020, the number of water and cultivated land patches increased, and the number of other types of plates decreased rapidly; this directly affects the fragmentation and separation of the landscape. Various types of landscapes tend to develop fragmented, the degree of concentration decreases and the internal stability decreases. Affected by lake inundation, woodland, grassland and bare land tend to be fragmented, the degree of concentration decreases, the anti-interference ability decreases and the vulnerability index increases. The water tends to develop contiguously, the degree of aggregation increases, the internal stability increases and the vulnerability of the landscape decreases.

**Table 3.** Calculation results of landscape pattern index for different land use types.

| Index | Years | Arable Land | Shrub | Woodland | Grass Land | Wetland | Water | Artificial Land | Bare Land |
|---|---|---|---|---|---|---|---|---|---|
| $A_i$ | 2000 | 73 | 521 | 212,634 | 33,371 | 37,423 | 2,436,421 | 19 | 213,973 |
| | 2010 | 72 | 244 | 120,946 | 18,999 | 39,949 | 2,516,975 | 20 | 98,527 |
| | 2020 | 374 | 39 | 71,921 | 8276 | 19,610 | 2,636,694 | 16 | 49,175 |
| $V_i$ | 2000 | 0.0356 | 2.6707 | 6.8142 | 10.7981 | 1.1275 | 2.9522 | 0.0048 | 3.1386 |
| | 2010 | 0.0210 | 2.6257 | 5.4648 | 11.2247 | 0.8330 | 1.5194 | 0.0481 | 2.9335 |
| | 2020 | 0.0200 | 0.7909 | 7.6371 | 11.8790 | 1.1190 | 1.0760 | 0.0085 | 3.8429 |
| $C_i$ | 2000 | 0.6400 | 2.9237 | 0.1453 | 0.8198 | 0.0493 | 0.0056 | 0.4096 | 0.0584 |
| | 2010 | 0.4981 | 2.0094 | 0.1891 | 0.9325 | 0.0659 | 0.0069 | 0.4425 | 0.1069 |
| | 2020 | 0.1603 | 4.4647 | 0.3137 | 1.2213 | 0.1245 | 0.0064 | 1.0856 | 0.1988 |
| $D_i$ | 2000 | 0.0020 | 0.0330 | 0.5927 | 0.4436 | 0.0555 | 0.5942 | 0.0004 | 0.2786 |
| | 2010 | 0.0014 | 0.0128 | 0.5255 | 0.3558 | 0.0750 | 0.7724 | 0.0004 | 0.2568 |
| | 2020 | 0.0025 | 0.0062 | 0.5743 | 0.2473 | 0.0681 | 0.8470 | 0.0006 | 0.2539 |
| $S_i$ | 2000 | 16.3112 | 14.2977 | 0.2696 | 1.2940 | 0.2274 | 0.1801 | 25.0136 | 0.1505 |
| | 2010 | 14.1308 | 16.1684 | 0.3495 | 1.6654 | 0.2555 | 0.2052 | 24.8827 | 0.2521 |
| | 2020 | 3.5337 | 58.4072 | 0.5524 | 2.6574 | 0.4889 | 0.2169 | 44.4901 | 0.4566 |
| $F_i$ | 2000 | 0.5802 | 38.1850 | 1.8373 | 13.9722 | 0.2564 | 0.5317 | 0.1192 | 0.4725 |
| | 2010 | 0.5016 | 1.8525 | 12.6419 | 11.1279 | 4.2899 | 11.0070 | 0.2508 | 8.5789 |
| | 2020 | 0.6465 | 1.1072 | 12.5351 | 8.3903 | 4.1234 | 10.8026 | 0.3441 | 8.2510 |

### 3.4.2. Ecological Risk Analysis of Typical Lake-Inundated Areas

From the spatial dimension (Figure 9), the ecological risks of lakes in different regions are quite different. The lower-risk areas in the eastern part of the Tibetan Plateau accounted for a large proportion. From 2000 to 2020, the lower-risk areas declined the most from 90% to 66.2%, the lower-risk areas were transformed into medium-high-risk areas, the area of the medium-risk area increased from 1.18% to 13.2% and medium-risk and higher-risk areas increased in the western region. In the northern region, the inundation area increased, resulting in a large change in the landscape type of the region. The reduction of lower and lowest risk decreased from 84.3% to 67.9%, and the higher-risk inundation area increased by 11%. The southern region is dominated by lower-risk areas. In 2000, there was no significant difference in the proportions of lower-risk areas, medium-risk areas and higher-risk areas, accounting for about 8% of the area, respectively. In 2020, the lower-medium-risk areas increased from 8.4% and 8.8% to 21.4% and 21.6%, and there was no significant change in the higher-risk and highest-risk areas.

From the time dimension (Figure 10), water is mainly distributed in lower-risk areas and lowest-risk areas from 2000 to 2020, mainly due to the small disturbance and vulnerability, and the improvement of the internal stability of the system. Cultivated land, shrubs and artificial land are mainly transferred to highest-risk areas, mainly due to the fragmentation and disturbance of the landscape being large, the interior of the system being fragile and the ecological risk rising. Bare land and wetlands are mainly distributed in medium-lower-risk areas, the increase in the medium-risk area is mainly due to the concentrated distribution of such landscapes, the reduced landscape vulnerability, and the relatively stable interior of the system. The landscape area of grassland and woodland decreased rapidly, the disturbance and vulnerability of the landscape increased and the landscape mainly shifted from medium-lower risk to medium-higher risk. The ecological risk level has increased as a whole, and the medium-risk and lower-risk areas accounted for the largest proportion in the study area. From 2000 to 2020, most areas were transformed from lower-lowest risk to medium-highest risk, and the lower-risk area declined most significantly, with a decline area of 3873.97 km$^2$; there was no major change in medium-risk areas from 2000 to 2010, and it increased rapidly by 2338.25 km$^2$ from 2010 to 2020; the higher-risk and highest-risk areas increased by 477.139 km$^2$.

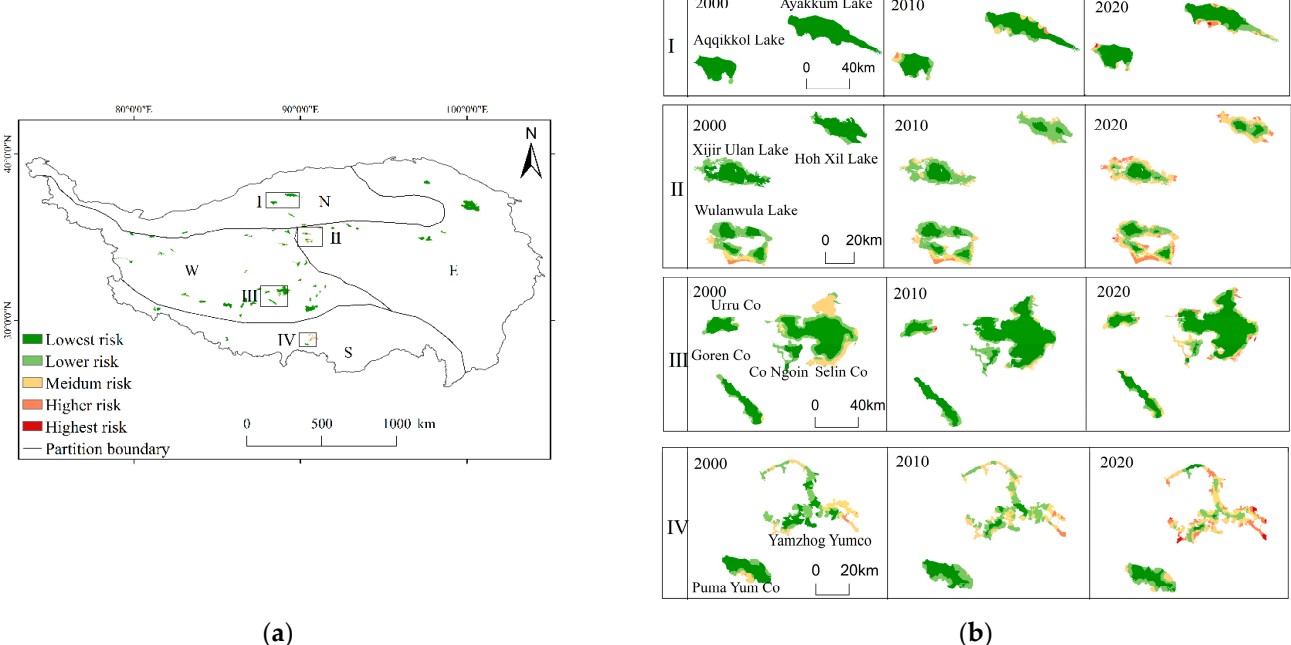

(**a**)　　　　　　　　　　　　　　　　　　　(**b**)

**Figure 9.** Landscape ecological risk of lake-inundated area level distribution. (**a**) Distribution of ecological risk levels in the Tibetan Plateau; (**b**) Spatial and temporal distribution of ecological risks in specific regions.

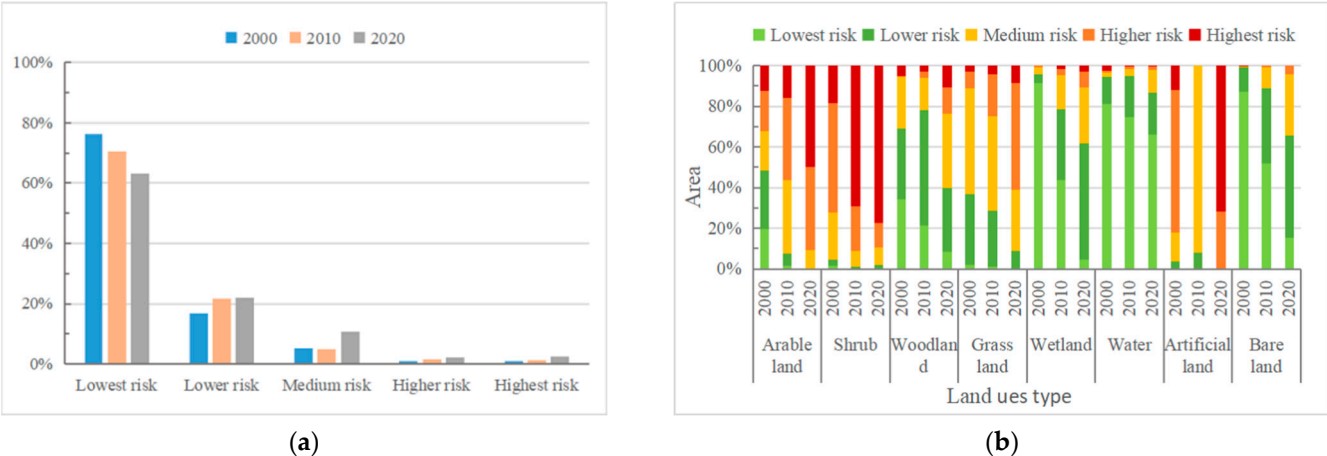

(**a**)　　　　　　　　　　　　　　　　　　　(**b**)

**Figure 10.** Area proportion of different landscape ecological risk. (**a**) Changes in the area of ecological risk areas; (**b**) Area changes of different landscape ecological risks.

## 4. Discussion

### 4.1. The Importance of Continuous Monitoring and Evaluation of Inundated Areas

In recent years, the results of dynamic monitoring of lakes by many scholars have shown that climate change on the Tibetan Plateau has led to massive melting of glaciers, sufficient precipitation and surface runoff replenishment and other comprehensive factors, resulting in lake expansion and lake inundation areas [66]. Most scholars discuss the impact of the expansion of a single lake, and there are few studies on the lake inundation area of the Tibetan Plateau [14,15].

Through the analysis of the research results of this paper, in terms of the extraction of the inundation area, the inundation area of the lake is compared with the lake change area obtained by Zhang et al. [67]. As shown in Figure 11, the four sets of data ($R^2_{2000-2005} = 0.8670$, $R^2_{2006-2010} = 0.8231$, $R^2_{2011-2015} = 0.8001$, $R^2_{2016-2020} = 0.8173$) show

good consistency. The extracted inundation area is larger than the lake expansion area. Using the inundation frequency index to calculate the inundation area can exert more accurate results, which provides a good case for the extraction of the lake inundation area. In terms of lake changes, the area increased linearly from 2000 to 2020, and, among them, salt and saltwater lakes had the largest expansion and lakes with glacier water as the main source of recharge expanded significantly, which is consistent with the conclusions of Lv et al. [43,47].

As shown in Figure 12, this study focuses on the dynamic monitoring of the lake inundation area. The change of the inundated area is different from the change trend of the lake water. By only studying the change of the lake area, we cannot fully understand the changing characteristics of the lake's hydrological fluctuation. The hydrological fluctuations make the lake-inundated area alternate between dry and wet, and the energy exchange is frequent. This area is more abundant in ecological functions than permanent water [17,58–69]. In recent years, the opening of remote sensing time series data and the availability of the GEE platform have provided the possibility to quickly and accurately obtain the extent of the inundation area. This paper provides a case study of long-term dynamic monitoring of the inundation area, which will greatly reduce the ground investigation work. Climate change brings uncertainty to lake inundation and brings new challenges to conservation management. In the later stage, changes in lake inundation frequency and surrounding landscape changes can be further predicted to obtain appropriate lake management indicators [68,70].

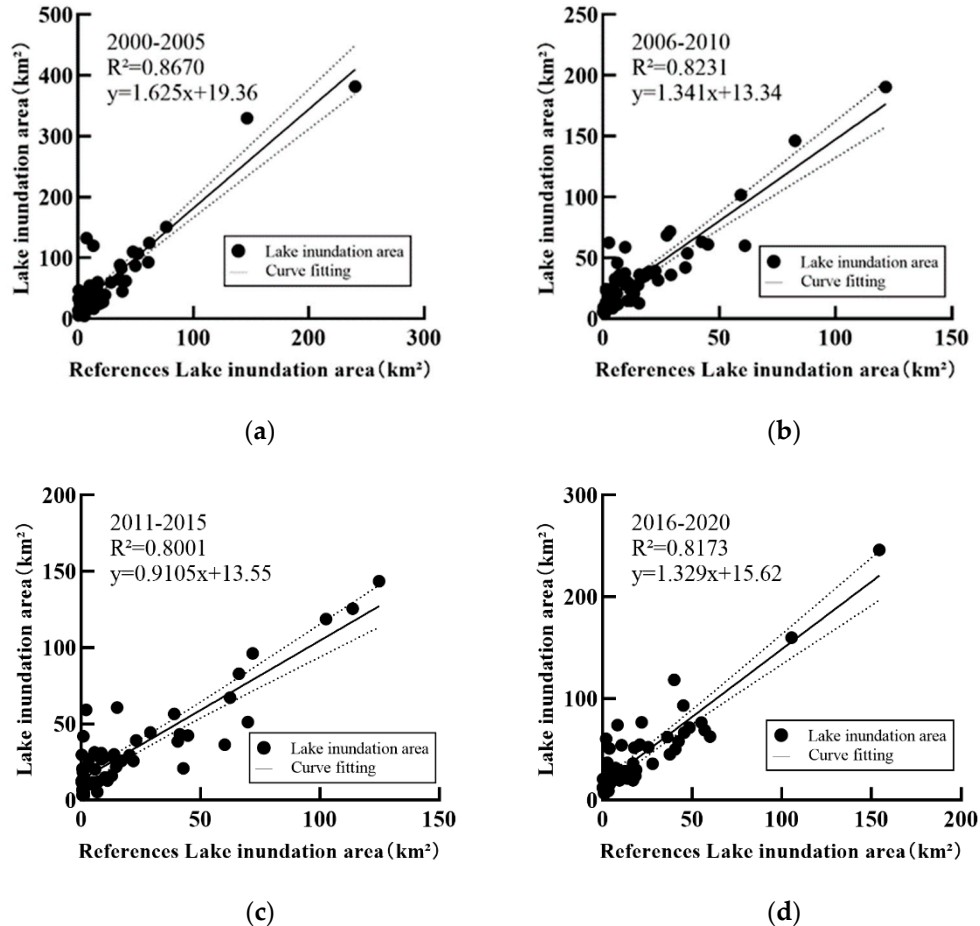

**Figure 11.** Comparison of the 52 lakes in this study and the reference results in the Tibetan Plateau. (**a**) 2000–2005; (**b**) 2006–2010; (**c**) 2011–2015; (**d**) 2016–2020.

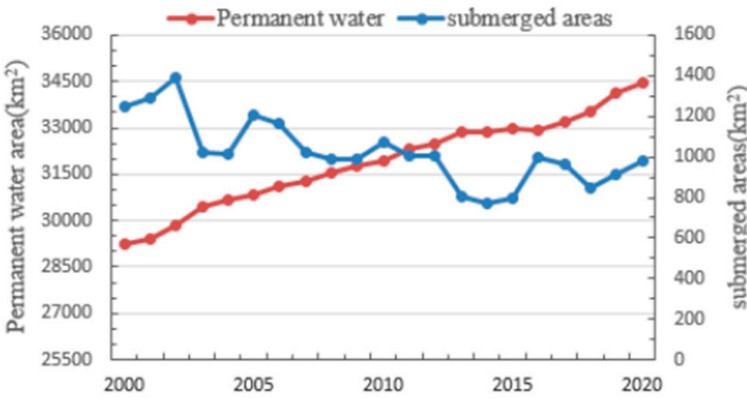

**Figure 12.** Hydrological changes of lakes on the Tibetan Plateau.

*4.2. Exploration of Ecological Risks in Inundated Areas*

This paper has made some breakthroughs in research methods. Landscape vulnerability in the construction of the landscape ecological risk index is mostly based on the expert scoring method to quantify landscape vulnerability as a constant. On the one hand, it is highly subjective, which has a certain impact on the objectivity of the results [37,38]. On the other hand, it cannot fully reflect the long-term dynamic changes of landscape vulnerability. According to the research results, from 2000 to 2020, the lake water and inundation frequency changed significantly, and the landscape changed greatly. The static landscape vulnerability index selection ignores the dynamic process of ecosystem evolution. This paper takes the lakes on the Tibetan Plateau as the research object, and combines the particularity of the study area to construct a landscape submergence frequency index, discussing the dynamic changes of its inundated area and the ecological risk analysis of lake-inundated areas; the results fully reflect the temporal and spatial heterogeneity of risks and provide a new idea for landscape ecological risk assessment. It is a powerful supplement to the existing dynamic monitoring of ecological risks in plateau lakes and makes up for the shortcomings of existing research to a certain extent.

Gu [47] research shows that the change of flood frequency will cause a series of ecological and environmental problems. Inundated areas will continue to increase, affecting the types and growth conditions of landscapes that survive in the arid environment, and the frequent alternating evolution of water and land makes some bare land transform into wetland and grassland, showing a positive evolution trend. The advantages and disadvantages of inundation frequency on different landscapes need further discussion and targeted governance. The ecological risk changes in the lake inundation area with natural hydrological fluctuations are quite heterogeneous, and the stability of the lake ecosystem can be maintained by human intervention in the later stage [71]. The protection of existing landscape ecology should be strengthened in lower-risk areas, and ecological restoration efforts should be further strengthened in medium-high-risk areas to prevent the transformation of medium-risk areas to higher-risk areas. The ecological risks of different inundated areas are significantly different, highlighting the necessity of implementing spatially differentiated risk management and long-term dynamic monitoring.

**5. Conclusions**

Based on long-term water data, this paper conducts dynamic monitoring of lake-inundated areas from four aspects; on the basis of it, ecological risk analysis was carried out, and the submergence frequency index was constructed as a dynamic index, which was applied to quantify the vulnerability of the landscape and quantitatively explore the temporal and spatial variation characteristics of ecological risk in lake-inundated areas. The main conclusions are as follows:

1. From 2000 to 2020, the inundated area of lakes first decreased and then increased, with the largest increase in 2002 and 2010. The lake inundation area is analyzed from four

aspects. Different lake inundation areas have different characteristics, and the northern region has the largest inundated area, accounting for 52.8% of the total inundated area; compared with saltwater lakes and freshwater lakes, salt lakes have the largest inundated area, reaching 156.85 km$^2$; the inundated area in low-altitude areas accounted for the largest proportion of the total inundated area, accounting for 55.6%; the area of lakes replenished by ice-snow melt water increased rapidly.

2. The change intensity of lake submergence frequency is relatively large from 2000 to 2020, indicating that the alternate changes of the land–water interface were enhanced, and the area of permanent lakes had increased. The lake area in the northern and western region of the Tibetan Plateau increased and the alternate changes of the inundated regions were enhanced, while the lake water in the southern region was relatively stable.

3. In the inundated area of the Tibetan Plateau, the area of bare land and grassland landscape change is the largest, which is mainly converted to water and wetland. In 2000, various landscapes (except water and cultivated land) were evenly distributed in various inundated areas; from 2010 to 2020, the landscape area of high-frequency inundation areas decreased rapidly, and the landscape area of medium-lower-frequency inundation areas increased.

4. In the inundated area of the Tibetan Plateau, the ecological risk level has risen. From 2000 to 2020, medium-risk and lower-risk areas accounted for the largest proportion in the study area, and most areas were transformed from lower-lowest risk to medium-highest risk. Among them, the areas of lower-, medium-, higher- and highest-risk areas increased by 4.97%, 5.52%, 1.22% and 1.49%, respectively. The area of the low ecological risk area decreased by 13.2%, and the medium-high ecological risk area showed a cluster-like change in space, showing a trend of gradually expanding to the outer layer. The ecological risk of cultivated land, shrub and artificial land is high, and the cultivated land, bare land and wetland distribution areas are generally under less ecological stress.

In this study, the lake ecosystem was divided into two parts, the inundated area and the permanent water, and the dynamic monitoring of the lake inundation area and inundation frequency was carried out, and the changing characteristics of the lake hydrological fluctuation are more comprehensively understood. On this basis, it quantitatively evaluates the landscape ecological risk in this area. Compared with previous studies, the construction of the landscape ecological risk index has been improved to a certain extent. The landscape inundation frequency index is applied to the dynamic quantification of the landscape vulnerability index, which provides a new perspective for the evaluation of landscape ecological risk. In future research, lakes with higher ecological risks can be selected for detailed analysis. The study area is specific to a lake and its surrounding areas, and the impact of lake inundation on the ecological environment, human production and life around the lake is further explored. Additionally, predictions can be made according to the development law of the inundated area to provide scientific support for preventing inundation risks.

**Author Contributions:** Conceptualization, D.W. and H.C.; Data Management, H.C., A.W. and S.W.; Formal Analysis, D.W. and H.C.; Capital Acquisition, D.W.; Survey, J.G.; Methodology, H.C. and Z.W.; Project Management, D.W.; Resources, D.W.; Software, K.W.; Supervision, H.W.; Verification, H.W.; Visualization, D.W., H.C., Z.W. and H.L.; Writing—Original Draft, H.C.; Writing—Review and Editing, A.W. and S.W. All authors have read and agreed to the published version of the manuscript.

**Funding:** This research was funded by the Second Tibetan Plateau of Scientific Expedition and Research Program (STEP), grant number 2019QZKK0608.

**Institutional Review Board Statement:** Not applicable.

**Informed Consent Statement:** Not applicable.

**Data Availability Statement:** The data presented in this study are available from the corresponding author on reasonable request.

**Acknowledgments:** The authors acknowledge Institute of Tibetan Plateau Research, Chinese Academy of Sciences for providing basic geographic and thematic data of Tibetan Plateau. The authors would also like to thank the editors and anonymous reviewers for their thorough and valuable comments and suggestions.

**Conflicts of Interest:** The authors declare no conflict of interest.

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
