# Peer review of "Dynamic Monitoring and Ecological Risk Analysis of Lake Inundation Areas in Tibetan Plateau"

_sustainability, doi:10.3390/su142013332_

Round 1

Reviewer 2 Report

Dear authors, 

It was my pleasure to review your work - I recently (two months ago) finished my PhD and this was my first academic review.

I think you have done a lot of hard work, but I unfortunately cannot recommend publishing this work without major revisions. My main reasons are as follows:

1) I have noted in the PDF (see my comments in text) many definitions that are lacking precision; 2) the word "quantitative" is used many times, but, in my opinion, some of the basic quantitative tools available for hydrologic change analysis, e.g. trend detection, which have been known to scientists since the 1940s, and used extensively in the field of hydrologic change since the 1980s are not employed; 3) many qualitative statements are made that are vague, e.g. the term "significant" is used in the conclusion without its typical statistical connotation; 4) the title is about ecological risk, but there is really no ecological data that supports your statements about vulnerability (and there is no discussion about what risk really is).

If you are to build extend this work to submit to any journal (or resubmit to the same journal), please consider exploring the literature on the following topics, in order to make the work stronger:

1) Trend detection: As a first step, please read the reports and papers by Kundzewicz which are a friendly introduction to the topic. For example, the report by Kundzewicz, Z.W. and Robson, A. (2000) Detecting Trend and Other Changes in Hydrological Data. World Meteorological Organization, Geneva, 157 and a paper by the same authors in an IAHS journal. Secondly, the papers by Yue & Wang (2002 & 2004) and Hamed and Rao (1998) maybe a good second step. For the scale of analysis you have employed, resampling (bootstrap) methods may be the acceptable standard for publication in good journals.

2) Please also consider reading the literature on IDF curves. There are numerous papers in the engineering hydrology literature (papers and books). 

However, if you are not interested (or comfortable) to go down the path of trend detection and IDF curves, another option (that does not make you veer too far from your current paper in terms of complexity of methods) is to consider incorporating some ecological data to strengthen your work and derive conclusions that are useful.

I regret to have to reject your paper at this time, but I hope you take my comments/suggestions/critique useful. I wish you much success in further development of this work!

You are welcome to contact me for additional literature pointers for either of the options.

Reviewer 3 Report

The recent climatic changes have globally changed the world biodiversity and pose a severe threat to the earth's ecosystem in general and the higher altitudes area, especially the mountains region in particular. The major change due to climate is the melting of the ice cap that, importantly, changes the lake inundation in the adjacent areas, which not only fluctuates the submerging area but also changes the diversity posing a severe ecological threat. The current paper address a well and important issue of the recent world problem. The idea and model used are wonderful and may attract broader readers. However, despite the importance of the idea, the manuscript language in most section is confused and seems very poor, although I am not native to English. Therefore, I highlighted some changes in different sections of the manuscript, which may further aid in the quality of the manuscript. In addition, I will recommend major revision with thorough text filtration for improvement.

Abstract

The abstract length is sufficient but it’s very hard to understand and came to some conclusion from the abstract. It is because the abstract lack proper method and don’t mention results. Therefore, the abstract needs thorough revisions for clear understanding of the readers.

- Line 14-17. A complicated long sentence that need thorough revision for meaningful information and understanding

-Line 17. The conclusions look very strange where the results are

-Line 27. Look out the first keyword please

Introduction

The introduction is to the point but need language improvement in the sentence structure and also looks out for topographic mistake

-Line 32, “flooding reflects the length and frequency of water flooding” better to use movement,

-Line 34-37, rephrase please

-Line 48, Jing Zhou [12] et al. Quantifies. Correct it please

-Line 53, remove “of them”

-Line 53, Cheng Jian [13] et al... is it journal format, check please if not please correct and also check other places

-Line 54, “other studies”, be specific please

Line 57 and 76, “many scholars “very non-scientific please avoid such writing

-Line 63-65, citation please

-Line 72, and “nature reserve” please add and

-Line 79, second time feel like objective of the paper, 1st time line 60-62, better to combine these two in the last paragraph of the introduction

-Line 94-94, revise please for clarity

-Study area part of introduction, would it not be better to move to materials and methods as objective formulation usually end the introduction section

Line125-132, seems parts of methodology

-No proper hypothesis and research objectives

Material and method

Well and comprehensive written

-No need of heading 3.1 and 3.2, please specify some other good heading

-Line 116, JRC?? Please write full form at first mention

-Line 118-119, Justify why 2000-2020, and also rephrased and used past tense please and follow in the whole methodology please

-Line 136-138, Rephrased please

-Line 163-165, please rephrased

-Line 178, the size????

-Line 179, “the paper” again and again using such redundant words is not suitable please change it

-Line 203, References [52]???????

-Line 209, sex. Very careless writing

-Line 216-221, rephrase please

Results

-Figure 2, the map missing the latitude an longitude please add

 -1, 2, 3 and 4 not needed in the paragraph of section 4.1

-Figure 3, please specify percent area meaning

-Figure 3a. The difference in numerical values is very high but it’s not visible on the map as such

-Line 251; please correct at the end after full stop and before 34.7%

-Line 254, please correct the punctuation

-Line 269-270 very much like part of discussion, better to delete from here and place in relevant area in discussion

-Figure 5, please add caption for a, b, c , d

-Line 303, Figure 7 or Figure 6 please correct

-Line 305, about 85.6%… please write the correct figure don’t mention approximations

-Please cite table 1 in the text

Discussion

Well written although a bit short, need to add more literature related to your work

Conclusion

-Line 406-411, No need of results repetition in conclusion

-Line 412-421, repetition of results, please conclude and evaluate for further studies

The conclusion is the repetition of the results. The conclusion needs to be refined and provide recommendations for further studies.

Reviewer 4 Report

The manuscript has analyzed and investigated the ecological risks of lake dynamics in the Lake Inundation. The manuscript is original and well put together. Majority of the results in the manuscript seem to be not validated, methodology is also not clear. Manuscript needs substantial modifications in its structure.

Round 2

Reviewer 1 Report

I would like to congratulate the Authors on their paper. They did a lot of hard work making corrections. The text and graphical part of the manuscript have been greatly improved. I recommend publishing it in the Sustainability.

Author Response

     We would like to thank you for your valuable comments on our paper. Your insightful review has enhanced our paper considerably.

Reviewer 3 Report

The authors have incorporated all the suggestions appropriately and are now suitable for publication in the current format.

Thank you

Author Response

(The authors gave the same response as above.)

Reviewer 4 Report

The problem of the manuscript is in the validation part. Unfortunately, its been claimed in the cover letter that authors have numerically assessed their results, but there is no clear description of how they have collected validation samples.

It is not clear whether samples are from visual interpretations or field visits/in-situ measurements? Just expressing the kappa coefficient value in the manuscript is not enough.

Further, other parts in this manuscript need to be validated (not only a classification evaluation by kappa coefficients). The paper in the current format (as-is) will be considered a numerical report with no validations. Authors have not precisely compared their results with other similar articles (as mentioned in the introduction, there are other similar scholars in the literature). This issue was mentioned in the first round of the review process as well.

In addition, the contribution and novelty of the paper are not well-expressed in the introduction.
